# Disruption of the Corticospinal Tract in Patients with Acute Lymphoblastic Leukemia: A Case Series

**DOI:** 10.3390/children9081223

**Published:** 2022-08-12

**Authors:** Jae Min Lee, Jong Bum Kim, Dong Hyun Byun, Su Min Son

**Affiliations:** 1Department of Pediatric Medicine, College of Medicine, Yeungnam University, Daegu 42415, Korea; 2Department of Physical Medicine and Rehabilitation, College of Medicine, Yeungnam University, Daegu 42415, Korea

**Keywords:** acute lymphoblastic leukemia, chemotherapy, motor, hemiplegia, diffusion tensor, corticospinal tract

## Abstract

Three patients who exhibited hemiplegic symptoms on conventional brain magnetic resonance imaging (MRI), during maintenance treatment for acute lymphoblastic leukemia, are reported. All patients exhibited unilateral motor weakness and poor hand function during chemotherapy. Conventional MRI revealed no definite abnormal lesions. However, in diffusion tensor tractography, the affected corticospinal tract on the contralateral side, consistently with clinical dysfunction, revealed disrupted integrity, decreased fractional anisotropy, and increased apparent diffusion coefficient compared to the results of the unaffected side or control participants. Control participants matched for age, sex, and duration from leukemia diagnosis, who underwent chemotherapy but had no motor impairments, exhibited preserved integrity of both corticospinal tracts. Diffusion tensor tractography can help evaluate patients with acute lymphoblastic leukemia and neurological dysfunction.

## 1. Introduction

Acute lymphoblastic leukemia (ALL) is a common type of childhood cancer. The survival rate for ALL has improved significantly to 75–85%, owing to the use of various treatment modalities, including chemotherapy, radiation therapy, and stem cell therapy [1,2,3]. However, it has also been reported that many survivors experience various neurological complications related to the central nervous system (CNS), including seizures, encephalopathy, or impairment of cognitive language function after treatment for ALL [4,5,6,7]. Previous studies have also investigated ALL survivors with motor dysfunction [8,9] or CNS chemotoxicity, even during treatment of ALL [10,11,12].

Detailed clinical evaluation of pediatric patients is often difficult. Children with disease, such as cancer, myopathy, or anemia, often exhibit poor physical condition, due to the characteristics of the disease, thus limiting clinical evaluation. In addition, chemotherapy sessions can interfere with a patient’s condition and sometimes cause poor compliance in pediatric patients with cancer. This delay in active diagnosis can lead to the failure of active and focused therapy, resulting in neurological sequelae after treatment, regardless of survival from leukemia. Therefore, the ability to predict and/or assess various symptoms during the treatment phase and overcome the limitations of clinical evaluation is emerging. 

Diffusion tensor tractography (DTT), derived from diffusion tensor imaging (DTI), can enable non-invasive visualization of the state of neural tracts at the subcortical level in the human brain in three dimensions and confirm the causal relationship between clinical symptoms and CNS origin [13,14]. Therefore, DTT can be useful when it is difficult to determine the cause of a patient’s clinical symptoms using conventional brain magnetic resonance imaging (MRI) [15,16,17].

In the present study, we report three cases of childhood ALL with hemiplegic symptoms consistent with DTT results.

## 2. Case Presentation

None of the three patients exhibited definite abnormal findings on conventional brain MRI, including T1- and T2-weighted and fluid-attenuated inversion recovery images, and these did not explain the hemiplegic symptoms. DTI, including diffusion-weighted imaging (DWI), apparent diffusion coefficient, fractional anisotropy (FA) images, and FA color imaging were performed. DTI data were acquired using a synergy-L Sensitivity Encoding (SENSE) six-channel head coil on a 1.5-T Philips Gyroscan Intera system (Hoffmann-La Roche, Best, The Netherlands). Sixty-seven contiguous slices, parallel to the anterior–posterior commissure line, were acquired. The imaging parameters were as follows: matrix, 128 × 128; field of view. 221 mm × 221 mm; echo time, 76 ms; repetition time, 10,726 ms; SENSE factor (parallel imaging reduction factor), 2; echo planar imaging factor, 67 and b = 1000 mm^2^ s^−1^; number of excitations, 1; and thickness, 2.3 mm (acquired voxel size 1.73 × 1.73 × 2.3 mm^3^).

DTI data were analyzed using the Oxford Center for Functional Magnetic Resonance Imaging of the Brain (FMRIB) Software Library (www.fmrib.ox.ac.uk/fsl accessed on 13 May 2020). Eddy current correction was applied to correct for the head motion effect and image distortion. FMRIB diffusion software with the routine options (0.5 mm step lengths, 5000 streamline samples, and curvature thresholds = 0.2) was used for fiber tracking. The pathway of the corticospinal tract (CST) was determined by the selection of fibers passing through seed and target (termination) regions of interest (ROIs). The CST was reconstructed by the selection of fibers passing through the ROI, as follows: between the CST portion of the pontomedullary junction (seed ROI) and CST portion of the anterior mid pons (target ROI) (Figure 1). 

Three pediatric patients diagnosed with ALL and exhibiting unilateral motor weakness and three age- and sex-matched, typically developing controls were recruited for this study. All control participants were diagnosed with ALL and underwent chemotherapy but did not experience any motor weakness or poor hand function during the treatment phase. All participants were diagnosed with precursor B-cell lymphoblastic leukemia without CNS involvement. Control participants did not exhibit any definite abnormal lesions on conventional brain MRI and no disrupted CSTs on DTT analysis. Demographic and functional data are summarized in Table 1 and Table 2.

### 2.1. Case 1

This patient was born at full term, with a birth weight of 3.7 kg, and no specific perinatal history. At six years of age, he was diagnosed with ALL and underwent chemotherapy induction and consolidation chemotherapy. During maintenance therapy, the patient complained of difficulty with left pinch grasp, 4 months after being diagnosed with ALL. In the clinical evaluation performed at that time, he exhibited left side motor weakness in his wrist and finger muscles, which was classified as a fair plus (3+) grade compared to the right side (4+). He also exhibited poor dexterity in his left finger on the right side (12 points) and left side (8 points) on the Purdue pegboard test, which is used to assess fine motor coordination. Conventional brain MRI did not reveal a definite abnormal lesion that could explain the left-side motor weakness. On DTT analysis, the right CST exhibited disrupted integrity, decreased fractional anisotropy (FA), and increased apparent diffusion coefficient (ADC) compared to the left CST, which was consistent with the patient’s left-side motor weakness (Figure 2) (Table 3).

### 2.2. Case 2

This patient was born at full term, with a birth weight of 3.6 kg, and no specific perinatal history. He was diagnosed with ALL at six years of age and underwent induction and consolidation chemotherapy. During the maintenance chemotherapy phase, 7 months after the diagnosis of ALL, he complained of right-side motor weakness and difficulty when drawing and using chopsticks despite his right-hand dominance. He exhibited poor dexterity (6 and 8 points on the right and left sides, respectively) on the Purdue pegboard test. In addition, he exhibited right-side motor weakness with a fair minus (3−) grade compared to the good minus (4−) grade of the left side at the elbow, wrist, and finger muscles. Conventional brain MRI revealed no definite abnormal lesions. DTT analysis revealed disrupted integrity in both CSTs; however, there was a prominent decrease in the left side compared to the right side, which was consistent with the patient’s right hemiplegic symptoms (Figure 2). Diffusion parameters of the left CST revealed decreased FA and increased ADC compared to the results of the right side or the results of the matched control participant (Table 3). 

### 2.3. Case 3

This patient was born at 37 ^+ 3^ weeks of gestational age, with a birth weight of 2.9 kg, and no specific perinatal history. She was diagnosed with ALL at 58 months of age, and underwent induction and consolidation chemotherapy, and also underwent subsequent chemotherapy reinduction and reconsolidation. She underwent maintenance chemotherapy, and 20 months after the diagnosis of ALL, exhibited left-side motor weakness and poor in-hand manipulation. She exhibited weakness in the left elbow, wrist, and finger muscles at a fair grade compared to the right (good) side. She exhibited poor dexterity (8 and 5 points on the right and left sides, respectively) on the Purdue pegboard test. Conventional brain MRI findings were non-specific. However, the DTT analysis revealed that the integrity of the right CST was disrupted (Figure 2), consistent with the patient’s left-side motor weakness. In addition, the right CST revealed decreased FA and increased ADC compared to the left side (Table 3).

## 3. Discussion

In this study, we performed DTT on three patients with ALL who exhibited hemiplegic symptoms during the treatment phase. We found disrupted integrity of the CST, decreased FA, and increased ADC on the side contralateral to the motor weakness, consistent with the clinical symptoms, although conventional MRI findings could not explain the patients’ motor symptoms. 

Aside from DTT, there are several modalities that can be used to assess brain function, including brain computed tomography, diffusion MRI, electroencephalography (EEG), magnetoencephalography (MEG), functional MRI, functional connectivity MRI, brain single-photon emission computed tomography (SPECT), position emission tomography (PET), and transcranial magnetic stimulation. DTT has a poor temporal resolution compared with EEG and MEG. However, DTT does not require the use of a contrast agent, unlike SPECT or PET, and involves less radiation compared to brain CT. In DTT, it is also possible to analyze various white matter tracts in one session. Chiefly, DTT is the only technology that can visualize white matter integrity at the subcortical level in vivo, in both adults and children. DTT can assess white matter tracts in three dimensions and can provide quantitative information about the condition of the white matter using diffusion parameters such as FA and ADC. After Mori et al. reported a CST study using DTT in 2003 [13], several DTT studies investigating the CST of patients with various pathologies, including stroke, traumatic brain injury, or cerebral palsy have been published [15,18,19]. The CST is known to control voluntary movements of the contralateral distal extremities. Therefore, disruptions in the CST are mainly related to the corresponding contralateral distal extremities, especially to the impairment of hand function [18]. Since the introduction of DTI, some studies have reported white matter injuries in patients with leukemia. Zou et al. performed DTI, to compare structural brain alterations in adult long-term survivors with a mean age of 40.71 years who were treated for ALL and in age-matched control participants. The authors reported that adult survivors exhibited reduced white matter volume compared with control participants in several brain regions [20]. Another DTI study by Aukema et al., involving patients who were treated for ALL with a mean age of 14 years and age-matched control participants, demonstrated that decreased white matter FA was correlated with motor speed in young childhood cancer survivors [21]. In 2014, Edelmann et al. performed DTI in long-term adult survivors of childhood ALL treated with chemotherapy only (mean age, 24.9 years), long-term survivors treated with cranial radiation therapy (mean age, 26.7 years), and healthy control participants (mean age, 23.1 years). They reported that survivors of ALL, regardless of treatment, exhibited a significantly low ratio of white matter to intracranial volume in the frontal and temporal lobes. They also reported significant neurocognitive impairments and a significant correlation between frontal and temporal brain volumes in adult survivors of ALL [22]. However, these studies compared brain volume or area between groups and did not identify corresponding neural tracts, such as the CST. In addition, these studies differed from ours, in that they targeted survivors, and not patients currently undergoing ALL treatment.

To the best of our knowledge, the present study was the first to investigate the CST in patients undergoing treatment for ALL. However, this study has some limitations, the first of which was its small sample size; as such, further studies involving a larger number of patients are necessary. Second, despite being a powerful anatomical imaging tool, DTI may underestimate or overestimate fiber tracts, because regions of fiber complexity and crossing can prevent full reflection of the underlying fiber architecture. 

In conclusion, this study investigated CST injuries in patients with hemiplegic symptoms and poor hand function following a diagnosis of ALL. We believe that DTT can be helpful in evaluating patients with leukemia and neurological symptoms. As such, further studies are warranted.

## Figures and Tables

**Figure 1 children-09-01223-f001:**
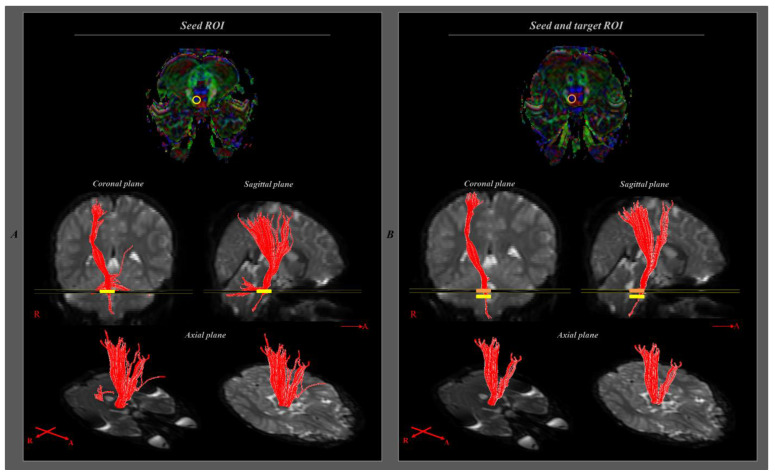
Diffusion tensor tractography tracking process. For the corticospinal tract analysis, the seed region of interest (ROI; yellow circle) was placed on the corticospinal tract portion of the pontomedullary junction, and the target ROI (orange circle) was positioned on the corticospinal tract portion of the anterior mid pons. When performing fiber tracking with one ROI (seed ROI), the transpontine fibers and pontocerebellar tract were also analyzed (**A**). However, the result of tracking with two ROIs (seed + target ROIs) revealed removed transpontine and pontocerebellar tracts, and only the corticospinal tract was analyzed (**B**).

**Figure 2 children-09-01223-f002:**
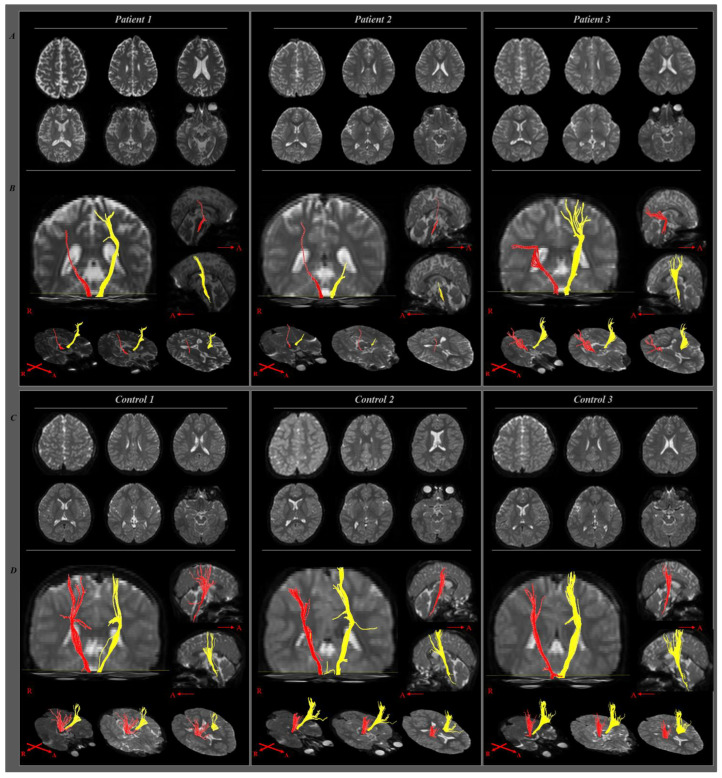
Diffusion tensor tractography of the corticospinal tract in patients and control participants. (**A**) and (**C**) show T2-weighted images of conventional magnetic resonance imaging, which reveal no definite abnormal findings. (**B**) and (**D**) show diffusion-tensor tractography of the corticospinal tract. Patients 1 and 3 exhibit disrupted integrity of the right corticospinal tract, and patient 2 exhibits disrupted integrity of the left corticospinal tract. Control participants have preserved integrities of both corticospinal tracts. Red indicates the right corticospinal tract and yellow indicates the left.

**Table 1 children-09-01223-t001:** Demographic and functional data of patients and control subjects.

	P1	P2	P3	C1	C2	C3
Age	6 y 4 m	6 y 2 m	4 y 10 m	6 y 3 m	6 y 3 m	4 y 11 m
Sex	Male	Male	Female	Male	Male	Female
Handedness	Right	Right	Right	Right	Right	Right
Period from diagnosis of ALL to examination	4 m	7 m	20 m	4 m	7 m	19 m
Purdue test	12/8	6/8	8/5	11/10	12/10	8/7

C1 was matched with P1, C2 was matched with P2, and C3 was matched with P3, respectively. P, patient; C, control; NS, non-specific. Y, years; M, months.

**Table 2 children-09-01223-t002:** Data on acute lymphoblastic leukemia diagnosis and treatment protocol.

	P1	P2	P3	C1	C2	C3
WBC at diagnosis (/µL)	2500	6860	2130	18,110	14,250	91,820
immunophenotype	SR	SR	SR	B	B + myeloid	SR
CNS status	negative	negative	negative	negative	negative	negative
Protocol	Modified BFM	Modified BFM	Modified BFM	Modified BFM	Modified BFM	Modified BFM
Risk group	SR	SR	SR	SR	SR	HR
Cytogenetics	Hyperdiploidy	Hyperdiploidy	Hyperdiploidy	Hyperdiploidy	TEL/AML1	MLL rearrangement

P, patient; C, control; WBC, white blood cell; SR, standard risk; HR, high risk; BFM, Berlin–Frankfurt–Munich. Risk groups were assigned by NCI criteria. High-risk (HR) patients were those with any of the following characteristics: age > 10 years, WBC > 50,000/µL, CNS disease, or unfavorable cytogenetics.

**Table 3 children-09-01223-t003:** Diffusion parameters of diffusion tensor tractography of patients and control subjects.

	P1	P2	P3	C1	C2	C3
FA affected *	0.412	0.225	0.415	0.587	0.562	0.515
Unaffected #	0.481	0.356	0.472	0.601	0.570	0.491
ADC affected *	0.96	1.144	0.95	0.84	0.769	0.921
unaffected #	0.901	0.927	0.94	0.867	0.801	0.894

C1 was matched with P1, C2 was matched with P2, and C3 was matched with P3, respectively. P, patient; C, control; FA, fractional anisotropy; ADC, apparent diffusion coefficient. * indicates the results of affected tract for patient or of right tract for control subject. # indicates the results of unaffected tract for patient or of left tract for control subject.

## Data Availability

All data are provided in the paper.

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
