# Peer review of "Disruption of the Corticospinal Tract in Patients with Acute Lymphoblastic Leukemia: A Case Series"

_children, 2022, doi:10.3390/children9081223_

Round 1
Reviewer 1 Report
Thank you for allowing me to review your paper. Below are some comments to improve it:
General:
-Title: It sounds like this is a study with a large number of patients, which is not true. I'd suggest shortening the title and including: "a case series".
-The entire manuscript warrants proofreading to improve English language usage. It might be confusing in some parts.
Abstract:
-Delete "unexplainable".
Introduction:
-Rephrase the first sentence, "childhood cancer" and "pediatric oncology" sounds redundant.
-Second paragraph, third sentence: I don't quite understand what you mean here, consider rephrasing or deleting it.
Case Presentation:
- First paragraph: combine the two first sentences.
- First paragraph: can you confirm the sequences included in conventional MRI? T1, T2, FLAIR? Also, confirm the planes - in the figures you're showing slices in the axial plane but you mentioned that "images were acquired parallel to the anterior-posterior commisure line", which is a structure that is mostly evaluated in the coronal plane.
- Did you explore other sequences such as DWI? ADC? Please comment and expand on that.
- Second paragraph: Cite the Software you used.
- Can you show a figure to illustrate the selection of ROIs and other post-processing steps?
- Can you include which chemotherapy regimen was indicated for each patient? For both cases and controls.
Discussion
-The discussion is somehow repetitive and hard to follow.
- Second paragraph: Can you mention which studies were performed in adults and which were done in children?
- Can you expand on the use of other imaging modalities/technologies to assess white matter integrity in children?
- What would be the clinical relevance of this study? Would you recommend DTI in all patients manifesting motor symptoms while undergoing chemotherapy for ALL? Would you recommend some other tests first?
Figure:
-What's the purpose of showing conventional axial images when compared to DTI? I'd suggest showing coronal T2 instead.
-Can you add labels to the colors in the figure's legend?
Table:
-Change age at diagnosis for years and months instead of months.
Author Response
Reviewer 1
Thank you for allowing me to review your paper. Below are some comments to improve it:
General:
-Title: It sounds like this is a study with a large number of patients, which is not true. I'd suggest shortening the title and including: "a case series".
Answer: I appreciate your comments for the manuscript. Following your comments, we revised the manuscript as follows.
Disruption of the Corticospinal Tract in Patients with Acute Lymphoblastic Leukemia: A Case Series
-The entire manuscript warrants proofreading to improve English language usage. It might be confusing in some parts.
Answer: Sorry about that. Editing was performed again by a professional expert.
Abstract:
-Delete "unexplainable".
Answer: Thank you for your comment. Following your comment, I delete ‘unexplainable’ and revised abstract.
Introduction:
-Rephrase the first sentence, "childhood cancer" and "pediatric oncology" sounds redundant.
Answer: I appreciate your comment for the manuscript. Following your comment, I revised the sentence.
-Second paragraph, third sentence: I don't quite understand what you mean here, consider rephrasing or deleting it.
Answer: Thank you for the comment. Following your comment, I revised the sentence.
Detailed clinical evaluation of pediatric patients is often difficult. Children with disease, such as cancer, myopathy, or anemia, often exhibit poor physical condition due to the characteristics of the disease, thus limiting clinical evaluation. In addition, chemotherapy sessions can interfere with a patient’s condition and sometimes cause poor compliance in pediatric patients with cancer. This delay in active diagnosis can lead to the failure of active and focused therapy, resulting in neurological sequelae after treatment, regardless of survival from leukemia.
Case Presentation:
- First paragraph: combine the two first sentences.
Answer: Thank you for your comment. In response to your opinion in this comment and the next comments, I revised the manuscript as follows.
None of the three patients exhibited definite abnormal findings on conventional brain MRI including T1- and T2- weighted images, and fluid-attenuated inversion recovery images, did not explain the hemiplegic symptoms. DTI, including diffusion-weighted imaging (DWI), apparent diffusion coefficient, fractional anisotropy (FA) images, and FA color imaging were performed.
- First paragraph: can you confirm the sequences included in conventional MRI? T1, T2, FLAIR?
Answer: I appreciate your valuable comment for the manuscript. Conventional MRI was performed with T1WI, T2WI, and FLAIR images, and DTI was taken with DWI, ADC, FA, and FA color images. I revised the manuscript following your comment.
None of the three patients exhibited definite abnormal findings on conventional brain MRI including T1- and T2- weighted images, and fluid-attenuated inversion recovery images, did not explain the hemiplegic symptoms. DTI, including diffusion-weighted imaging (DWI), apparent diffusion coefficient, fractional anisotropy (FA) images, and FA color imaging were performed.
- Also, confirm the planes - in the figures you're showing slices in the axial plane but you mentioned that "images were acquired parallel to the anterior-posterior commisure line", which is a structure that is mostly evaluated in the coronal plane.
Answer: Slicing parallel to the ant-post commissure line is used to secure to obtain anatomically identical axial-sectional result in the same subject. This is always done when taking all conventional MRIs as well as DTI in our hospital. If there does not exist the guideline about slicing, the axial image can changes each time it is taken, because of various position of participant’s neck flexion angle. If the anatomical brain structures that should be referenced in the imaging results are not the same, may be biased in the evaluation of brain lesion. So, I apply this criterion to all patients.
Unfortunately, coronal images were not obtained. We will refer to it for future research. I appreciate your valuable comment.
- Did you explore other sequences such as DWI? ADC? Please comment and expand on that.
Answer: I appreciate your comment. As mentioned in the previous question, conventional MRI was performed with T1WI, T2WI, and FLAIR images, and DTI was taken with DWI, ADC, FA, and FA color images. I revised the manuscript following your comment.
None of the three patients exhibited definite abnormal findings on conventional brain MRI including T1- and T2- weighted images, and fluid-attenuated inversion recovery images, did not explain the hemiplegic symptoms. DTI, including diffusion-weighted imaging (DWI), apparent diffusion coefficient, fractional anisotropy (FA) images, and FA color imaging were performed.
- Second paragraph: Cite the Software you used.
Answer: We revised the manuscript as follows.
DTI data were analyzed using the Oxford Center for Functional Magnetic Resonance Imaging of the Brain (FMRIB) Software Library (www.fmrib.ox.ac.uk/fsl). Eddy current correction was applied to correct for the head motion effect and image distortion.
- Can you show a figure to illustrate the selection of ROIs and other post-processing steps?
Answer: We added the figure about the selection of ROIs and fiber tracking process (Figure 1).
Figure 1. Diffusion tensor tractography tracking process. For the corticospinal tract analysis, the seed region of interest (ROI; yellow circle) was placed on the corticospinal tract portion of the pontomedullary junction, and the target ROI (orange circle) was positioned on the corticospinal tract portion of the anterior mid pons. When performing fiber tracking with one ROI (seed ROI), the transpontine fibers and pontocerebellar tract are also analyzed (A). However, the result of tracking with two ROIs (seed + target ROIs) reveals removed transpontine and pontocerebellar tracts, and only the corticospinal tract is analyzed (B).
- Can you include which chemotherapy regimen was indicated for each patient? For both cases and controls.
Answer: Thank you for your comment. I added Table 2 about ALL diagnosis and chemotherapy regimen.
Table 2. Data on Acute lymphoblastic leukemia diagnosis and treatment protocol
|
P1 |
P2 |
P3 |
C1 |
C2 |
C3 |
||
|
WBC at diagnosis (/uL) |
2500 |
6860 |
2130 |
18110 |
14250 |
91820 |
|
|
immunophenotype |
SR |
SR |
SR |
B |
B+myeloid |
SR |
|
|
CNS status |
negative |
negative |
negative |
negative |
negative |
negative |
|
|
Protocol |
Modified BFM |
Modified BFM |
Modified BFM |
Modified BFM |
Modified BFM |
Modified BFM |
|
|
Risk group |
SR |
SR |
SR |
SR |
SR |
HR |
|
|
Cytogenetics |
Hyperdiploidy |
Hyperdiploidy |
Hyperdiploidy |
Hyperdiploidy |
TEL/AML1 |
MLL rearrangement |
|
P, patient; C, control; WBC, white blood cell; SR, standard risk; HR, high risk; BFM, Berlin-Frankfurt-Munich
Risk groups were assigned by NCI criteria. High-risk (HR) patients were those with any of the following characteristics: age >10 years, WBC >50,000/uL, CNS disease, or unfavorable cytogenetics.
Discussion
-The discussion is somehow repetitive and hard to follow.
Answer: Sorry for that. To help CHILDREN’s readers understand, the manuscript was revised and edited again.
- Second paragraph: Can you mention which studies were performed in adults and which were done in children?
Answer: I revised manuscript in the discussion section for more detailed description following your comment.
Since the introduction of DTI, some studies have reported white matter injuries in patients with leukemia. Zou et al. performed DTI to compare structural brain alterations in adult long-term survivors with a mean age of 40.71 years who were treated for ALL and in age-matched control participants. The authors reported that adult survivors exhibited reduced white matter volume compared with control participants in several brain regions [20]. Another DTI study by Aukema et al. involving patients who were treated for ALL with a mean age of 14 years and age-matched control participants demonstrated that decreased white matter FA was correlated with motor speed in young childhood cancer survivors [21]. In 2014, Edelmann et al. performed DTI in long-term adult survivors of childhood ALL treated with chemotherapy only (mean age, 24.9 years), long-term survivors treated with cranial radiation therapy (mean age, 26.7 years), and healthy control participants (mean age, 23.1 years). They reported that survivors of ALL, regardless of treatment, exhibited a significantly low ratio of white matter to intracranial volume in the frontal and temporal lobes. They also reported significant neurocognitive impairments and a significant correlation between frontal and temporal brain volumes in adult survivors of ALL [22].
- Can you expand on the use of other imaging modalities/technologies to assess white matter integrity in children?
Answer: To the best of our knowledge, DTI is the only tool that can visualize white matter integrity at the subcortical level in vivo in adults as well as children. However, there are several technologies can be used for assessment of brain function such as EEG, MEG, fMRI, functional connectivity MRI, fNIRS, PET, SPECT. In addition, TMS can be used as electrophysiological evaluation. MRS can be used as metabolic imaging. Each modality has different strengths. That is, EEG and MEG have good temporal resolution but relatively weak spatial resolution, and DTI has good spatial resolution but poor temporal resolution.
Following your comment, we revised the manuscript in the discussion section.
In this study, we performed DTT on three patients with ALL who exhibited hemiplegic symptoms during the treatment phase. We found disrupted integrity of the CST, decreased FA, and increased ADC on the side contralateral to motor weakness, consistent with the clinical symptoms, although conventional MRI findings could not explain the patients’ motor symptoms.
Aside from DTT, there are several modalities that can be used to assess brain function including brain computed tomography, diffusion MRI, electroencephalography (EEG), magnetoencephalography (MEG), functional MRI, functional connectivity MRI, brain single-photon emission computed tomography (SPECT), position emission tomography (PET), and transcranial magnetic stimulation. DTT has poor temporal resolution compared with EEG and MEG. However, DTT does not require the use of a contrast agent compared to SPECT or PET, and involves less radiation compared to brain CT. In DTT, it is also possible to analyze various white matter tracts in one session. Chiefly, DTT is the only technology that can visualize white matter integrity at the subcortical level in vivo, in both adults and children. DTT can assess white matter tracts in three dimensions and can provide quantitative information about the condition of the white matter using diffusion parameters such as FA and ADC. After Mori et al. reported a CST study using DTT in 2003 [13], several DTT studies investigated the CST of patients with various pathologies, including stroke, traumatic brain injury, or cerebral palsy, have been published [15,18,19].
- What would be the clinical relevance of this study? Would you recommend DTI in all patients manifesting motor symptoms while undergoing chemotherapy for ALL? Would you recommend some other tests first?
Answer: Patients underwent chemotherapy for ALL often show various neurological symptoms, and MRI is performed when chemotherapy-induced encephalopathy is suspected. However, conventional MRI has limitations in providing detailed information on the white matter state correlated with neurological symptoms. The delay in accurate diagnosis leads to delay in treatment and affects the patient's clinical prognosis. Therefore, when MRI is considered, the additional conduct of DTT provides the basis for a detailed diagnosis of patients with ALL. The authors believe that the results of this case report reveal the clinical usefulness of DTT in ALL patients. In addition, it is possible to analyze various white matter tracts in one performance of DTI; CST, Medial Lemniscus, and corticoreticulospinal tract related to motor symptoms, fornix related to short term memory, cingulum related to high cognition and long term memory, arcuate fasciculus related to language function, thalamocortical pathway related to attention or impulsivity, ARAS related to arousal, corticobulcar tract related to dysphagia.
Therefore, if neurological symptoms appear in patients underwent chemotherapy for ALL, and chemotherapy-induced encephalopathy is suspected, the authors recommend possible clinical evaluations and DTI.
Figure:
-What's the purpose of showing conventional axial images when compared to DTI? I'd suggest showing coronal T2 instead.
Answer: I appreciate your comment. We added conventional axial images to the figure to show that the patients reveal clinical symptom of hemiplegia, but there is no lesion to explain their hemiplegia in conventional MRI.
Unfortunately, coronal T2 images were not obtained. In the future, when taking DTI in patients, we will take an additional coronal images according to your valuable comment.
Thank you again.
-Can you add labels to the colors in the figure's legend?
Answer: Sorry for that. I revised the figure by labeling to the colors.
Figure legend
Figure 2. Diffusion tensor tractography of the corticospinal tract in patients and control participants. A and C show T2-weighted images of conventional magnetic resonance imaging, which reveal no definite abnormal findings. B and D show diffusion-tensor tractography of the corticospinal tract. Patients 1 and 3 exhibit disrupted integrity of the right corticospinal tract, and patient 2 exhibits disrupted integrity of the left corticospinal tract. Control subjects have preserved integrities of both corticospinal tracts. Red color indicates right corticospinal tract, and yellow color indicates left.
Table:
-Change age at diagnosis for years and months instead of months.
Answer: I changed age notation to age and months as your comments. Thank you for your comment

Reviewer 2 Report
The authors describe the course of 3 children with ALL who developed (treatment-associated) hemiplegia. No abnormality was found on conventional structural MRI, whereas DTI sequences showed abnormalities. These findings were compared with 3 children with ALL without hemiplegia. On the figures of the MRIs, one can follow the authors' assessment in interrupted and non-interrupted tracts. However, in order to compare results in future studies, it would be useful to show information about the DTI parameters. At least the fractional anisotropy value and the apparent diffusion coefficient of the CST should be given and compared. This would be the basis to integrate DTI-based MRI sequences as a diagnostic tool in routine imaging in the future. The classification alone into interrupted and non-interrupted is too subjective to be applied in further studies.
Author Response
Reviewer 2
The authors describe the course of 3 children with ALL who developed (treatment-associated) hemiplegia. No abnormality was found on conventional structural MRI, whereas DTI sequences showed abnormalities. These findings were compared with 3 children with ALL without hemiplegia. On the figures of the MRIs, one can follow the authors' assessment in interrupted and non-interrupted tracts. However, in order to compare results in future studies, it would be useful to show information about the DTI parameters. At least the fractional anisotropy value and the apparent diffusion coefficient of the CST should be given and compared. This would be the basis to integrate DTI-based MRI sequences as a diagnostic tool in routine imaging in the future. The classification alone into interrupted and non-interrupted is too subjective to be applied in further studies.
Answer: I appreciate your valuable comment. I added the table 3 for the result of FA and ADC. I revised the manuscript as follows.
Table 3. Diffusion parameters of diffusion tensor tractgraphy of patients and control subjects
|
P1 |
P2 |
P3 |
C1 |
C2 |
C3 |
||
|
FA affected* |
0.412 |
0.225 |
0.415 |
0.587 |
0.562 |
0.515 |
|
|
Unaffected# |
0.481 |
0.356 |
0.472 |
0.601 |
0.570 |
0.491 |
|
|
ADC affected* |
0.96 |
1.144 |
0.95 |
0.84 |
0.769 |
0.921 |
|
|
unaffected# |
0.901 |
0.927 |
0.94 |
0.867 |
0.801 |
0.894 |
|
C1 was matched with P1, C2 was matched with P2, and C3 was matched with P3, respectively.
P, patient; C, control; FA, fractional anisotropy; ADC, apparent diffusion coefficient
* indicates the results of affected tract for patient or of right tract for control subject.
# indicates the results of unaffected tract for patient or of left tract for control subject.
Abstract: …Conventional MRI revealed no definite abnormal lesions. However, in diffusion tensor tractography, the affected corticospinal tract on he contralateral sideconsistent with clinical dysfunction revealed disrupted integrity, decreased fractional anisotropy and increased apparent diffusion coefficient compared to the results of the unaffected side or control participants.
2.1. Case 1
…His conventional brain MRI did not reveal a no definite abnormal lesion that could explain the left-side motor weakness. On DTT analysis, the right CST showed disrupted integrity, decreased fractional anisotropy (FA) and increased apparent diffusion coefficient (ADC) compared to the left CST, which was consistent with the patient’s left-side motor weakness (Figure 2) (Table 3).
2.2. Case 2
...Conventional brain MRI revealed no definite abnormal lesions. DTT analysis revealed disrupted integrities in both CSTs, howeve, a prominent decrease in the left side compared to the right side, which was consistent with the patient’s right hemiplegic symptoms (Figure 2). Diffusion parameters of left CST revealed decreased FA and increased ADC compared to the results of the right side or the results of the matched control participant (Table 3).
2.3. Case 3
…Conventional brain MRI findings were non-specific. However, the DTT analysis revealed that the integrity of the right CST was disrupted (Figure 2), consistent with the patient’s left-side motor weakness. In addition, right CST revealed decreased FA and increased ADC compared to the left side (Table 3).

Reviewer 3 Report
Dear authors,
Thank you for sharing your case report. I found it very intersting. Please find below a few comment, suggestions:
-Pg 1 ln 35: neurological evaluation is also frequently impaired by the clinical status of the patient (not only his cooperation): ie: myopathy, asthenia, anemia,... Could you please add this point?
- Table 1: could you please add more info about the diagnosis: B or T ALL with or without CNS involvement.
Cases reports 1-3: Could you please also give a bit more or info about treatment received (which protocol, low risk high risk ALL, simple vs triple IT)
- Methodology: was general anesthesia required for the acquisition of MRI?
- Discussion: could you discuss a bit further the advantages/disadvantages of DTT please? Cost effectiveness, timing of acquisition, availability?
Thank you again for sharing your work
Best wishes
Author Response
Reviewer 3
Dear authors,
Thank you for sharing your case report. I found it very interesting. Please find below a few comment, suggestions:
-Pg 1 ln 35: neurological evaluation is also frequently impaired by the clinical status of the patient (not only his cooperation): ie: myopathy, asthenia, anemia,... Could you please add this point?
Answer: I appreciate your valuable comment. I revised the introduction section as follows.
Detailed clinical evaluation of pediatric patients is often difficult. Children with disease, such as cancer, myopathy, or anemia, often exhibit poor physical condition due to the characteristics of the disease, thus limiting clinical evaluation. In addition, chemotherapy sessions can interfere with a patient’s condition and sometimes cause poor compliance in pediatric patients with cancer. This delay in active diagnosis can lead to the failure of active and focused therapy, resulting in neurological sequelae after treatment, regardless of survival from leukemia.
- Table 1: could you please add more info about the diagnosis: B or T ALL with or without CNS involvement.
Answer: Thank you for the comment. All participants were the patients were diagnosed as precursor B-cell lymphoblastic leukemia without CNS involvement. We added the information about the diagnosis.
Three pediatric patients diagnosed with ALL and exhibiting unilateral motor weakness and three age- and sex-matched, typically developing controls were recruited for this study. All control participants were diagnosed with ALL and underwent chemotherapy, but did not experience any motor weakness or poor hand function during the treatment phase. All participants were diagnosed with precursor B-cell lymphoblastic leukemia without CNS involvement. Control participants showed no definite abnormal lesions on conventional brain MRI and no disrupted CSTs on DTT analysis. Demographic and functional data are summarized in Table 1 and 2.
Cases reports 1-3: Could you please also give a bit more or info about treatment received (which protocol, low risk high risk ALL, simple vs triple IT)
Answer: Thank you for your comment. I added Table 2 about ALL diagnosis and treatment protocol.
Table 2. Data on Acute lymphoblastic leukemia diagnosis and treatment protocol
|
P1 |
P2 |
P3 |
C1 |
C2 |
C3 |
||
|
WBC at diagnosis (/uL) |
2500 |
6860 |
2130 |
18110 |
14250 |
91820 |
|
|
immunophenotype |
SR |
SR |
SR |
B |
B+myeloid |
SR |
|
|
CNS status |
negative |
negative |
negative |
negative |
negative |
negative |
|
|
Protocol |
Modified BFM |
Modified BFM |
Modified BFM |
Modified BFM |
Modified BFM |
Modified BFM |
|
|
Risk group |
SR |
SR |
SR |
SR |
SR |
HR |
|
|
Cytogenetics |
Hyperdiploidy |
Hyperdiploidy |
Hyperdiploidy |
Hyperdiploidy |
TEL/AML1 |
MLL rearrangement |
|
P, patient; C, control; WBC, white blood cell; SR, standard risk; HR, high risk; BFM, Berlin-Frankfurt-Munich
Risk groups were assigned by NCI criteria. High-risk (HR) patients were those with any of the following characteristics: age >10 years, WBC >50,000/uL, CNS disease, or unfavorable cytogenetics.
- Methodology: was general anesthesia required for the acquisition of MRI?
Answer: These patients underwent MRI in natural sleep state without anesthesia.
- Discussion: could you discuss a bit further the advantages/disadvantages of DTT please? Cost effectiveness, timing of acquisition, availability?
Answer: I appreciate your valuable comment. I revised the discussion section as follows.
In this study, we performed DTT on three patients with ALL who exhibited hemiplegic symptoms during the treatment phase. We found disrupted integrity of the CST, decreased FA, and increased ADC on the side contralateral to motor weakness, consistent with the clinical symptoms, although conventional MRI findings could not explain the patients’ motor symptoms.
Aside from DTT, there are several modalities that can be used to assess brain function including brain computed tomography, diffusion MRI, electroencephalography (EEG), magnetoencephalography (MEG), functional MRI, functional connectivity MRI, brain single-photon emission computed tomography (SPECT), position emission tomography (PET), and transcranial magnetic stimulation. DTT has poor temporal resolution compared with EEG and MEG. However, DTT does not require the use of a contrast agent compared to SPECT or PET, and involves less radiation compared to brain CT. In DTT, it is also possible to analyze various white matter tracts in one session. Chiefly, DTT is the only technology that can visualize white matter integrity at the subcortical level in vivo, in both adults and children. DTT can assess white matter tracts in three dimensions and can provide quantitative information about the condition of the white matter using diffusion parameters such as FA and ADC. After Mori et al. reported a CST study using DTT in 2003 [13], several DTT studies investigated the CST of patients with various pathologies, including stroke, traumatic brain injury, or cerebral palsy, have been published [15,18,19].

Round 2
Reviewer 1 Report
Dear authors,
Thank you for revising your manuscript as suggested. I think it has improved substantially and is suitable for publication.